# Homologous Recombination Repair Deficiency in Metastatic Prostate Cancer: New Therapeutic Opportunities

**DOI:** 10.3390/ijms25094624

**Published:** 2024-04-24

**Authors:** Claudia Piombino, Stefania Pipitone, Elena Tonni, Luciana Mastrodomenico, Marco Oltrecolli, Cyrielle Tchawa, Rossana Matranga, Sara Roccabruna, Elisa D’Agostino, Marta Pirola, Francesca Bacchelli, Cinzia Baldessari, Maria Cristina Baschieri, Massimo Dominici, Roberto Sabbatini, Maria Giuseppa Vitale

**Affiliations:** 1Division of Oncology, Department of Oncology and Hematology, University Hospital of Modena, 41124 Modena, Italy; 256171@studenti.unimore.it (C.P.); pipitone.stefania@aou.mo.it (S.P.); 325770@studenti.unimore.it (E.T.); 297001@studenti.unimore.it (L.M.); 297143@studenti.unimore.it (M.O.); 308106@studenti.unimore.it (C.T.); 297415@studenti.unimore.it (R.M.); 279224@studenti.unimore.it (S.R.); 297407@studenti.unimore.it (E.D.); 310546@studenti.unimore.it (M.P.); baldessari.cinzia@aou.mo.it (C.B.); massimo.dominici@unimore.it (M.D.); sabbrob@unimore.it (R.S.); 2Clinical Trials Office, Division of Oncology, Department of Medical and Surgical Sciences for Children & Adults, University of Modena and Reggio Emilia, 41124 Modena, Italy; francesca.bacchelli@unimore.it; 3Laboratory of Cellular Therapy, Division of Oncology, Department of Medical and Surgical Sciences for Children & Adults, University of Modena and Reggio Emilia, 41124 Modena, Italy; mariacristina.baschieri@unimore.it

**Keywords:** metastatic prostate cancer, homologous recombination repair, *ATM*, *BRCA1*, *BRCA2*, androgen receptor, PARP inhibitor

## Abstract

More than 20% of metastatic prostate cancer carries genomic defects involving DNA damage repair pathways, mainly in homologous recombination repair-related genes. The recent approval of olaparib has paved the way to precision medicine for the treatment of metastatic prostate cancer with PARP inhibitors in this subset of patients, especially in the case of *BRCA1* or *BRCA2* pathogenic/likely pathogenic variants. In face of this new therapeutic opportunity, many issues remain unsolved. This narrative review aims to describe the relationship between homologous recombination repair deficiency and prostate cancer, the techniques used to determine homologous recombination repair status in prostate cancer, the crosstalk between homologous recombination repair and the androgen receptor pathway, the current evidence on PARP inhibitors activity in metastatic prostate cancer also in homologous recombination repair-proficient tumors, as well as emerging mechanisms of resistance to PARP inhibitors. The possibility of combination therapies including a PARP inhibitor is an attractive option, and more robust data are awaited from ongoing phase II and phase III trials outlined in this manuscript.

## 1. Introduction

Metastatic prostate cancer (PC) is still associated with a dismal prognosis, being associated with a 5-year relative survival rate of only 34.1% [1]. The recent advent of precision medicine for the treatment of PC is slowly changing the prognosis of the subgroup of patients affected by metastatic disease carrying somatic or germline pathogenic/likely pathogenic variants (P/LPVs) in *BRCA1* or *BRCA2*. Metastatic PC has been traditionally considered an androgen-dependent tumor whose progression includes first a hormone-sensitive phase (mHSPC) and then a castration-resistant phase (mCRPC) associated with poor response to subsequent therapies [2]. However, PC is a clinically heterogeneous disease as demonstrated, for example, by the extremely variable duration of the hormone-sensitive phase due to intra- and inter-tumor different timings and the mechanisms of the onset of resistance to androgen deprivation therapy (ADT) [3]. The presence of various clinical phenotypes reflects a considerable molecular heterogeneity among patients [4].

Nearly 20% of primary PC carries genomic defects involving DNA damage repair (DDR) pathways: the most frequently P/LPVs occur in *BRCA2* (3%), *BRCA1* (1%), *CDK12* (2%), *ATM* (4%), *FANCD2* (7%), and *RAD51C* (3%) [4,5]. In mCRPC, the rate of somatic P/LPVs in DDR-associated genes increases up to 23%, mainly in *BRCA2* (13%), *ATM* (7.3%), and *MSH2* (2%) [5]. In metastatic PC, a germline P/LPV in DDR-associated genes is detectable in about 12% of patients, mainly in *BRCA2* (5.3%), *ATM* (1.6%), *CHEK2* (1.9%), *BRCA1* (0.9%), *RAD51D* (0.4%), *PALB2* (0.4%), and *ATR* (0.3%) [5]. Moreover, PC in patients carrying a germline P/LPV in *BRCA2* appears to occur earlier, has a more aggressive phenotype, a high risk of recurrence after surgery, and is associated with significantly reduced survival than non-carrier patients [6,7,8,9,10,11,12,13].

Most of the aforementioned genes encode for tumor suppressor proteins involved in homologous recombination repair (HRR), an error-free mechanism that repairs DNA double-strand breaks (DSBs). HRR deficiency (HRD) induces the activation of other error-prone DDR pathways; cells repaired via these mechanisms undergo complex genomic rearrangements and apoptosis [14,15]. The subgroup of patients affected by metastatic PC carrying a germline and/or somatic P/LPV in HRR-associated genes that determine a HRD is supposed to be responsive to poly-(ADP-ribose) polymerase (PARP) inhibitors (PARPis). PARPis act mainly by the inhibition of the catalytic activity of PARP and by trapping PARP at sites of DNA single-strand breaks (SSBs) [16,17]. The inhibition of the catalytic activity of PARP promotes SSBs which, if unrepaired, consequently lead to DSBs during DNA replication, resulting in synthetic lethality in cells with HRD [18,19,20]. Moreover, the action of PARPis binding to the catalytic domain of PARP allosterically modifies interactions between DNA and the DNA-binding domain of the protein, to the point that PARP becomes trapped on DNA [21] (Figure 1).

The approval of the PARPi olaparib as single agent for the treatment of HRR gene-mutated mCRPC has revolutionized the therapeutic scenario of metastatic PC, being the first target agent approved in PC [22]. Nevertheless, in face of this new therapeutic opportunity, many issues remain unsolved: whether PARPis are effective for all patients with HRD; how to determine HRD in PC; the role of PARPis in patients with HRR-proficient PC; how to overcome resistance to PARPis; and the future of precision medicine in metastatic PC. With the aim to give an answer to these questions, this narrative review describes the available data on the relationship between HRR and PC, the current role of PARPis in metastatic PC, as well as the ongoing phase II and phase III trials investigating PARPi-based therapeutic strategies in metastatic PC.

## 2. Overview on HRR Pathway

A brief description of the HRR pathway, with a focus on the function of each protein involved, is fundamental to understand what happens in tumor cells with HRD. In fact, genome integrity plays a fundamental role in tumorigenesis. Thus, whether caused by exogenous or endogenous stress, or even just by errors occurring during replication, DNA damage needs to be repaired to correctly carry genetic information on. According to the different kinds of damage, cells can rely on multiple DDR pathways. One of the most important is HRR [23,24,25,26].

HRR is a nearly error-free DNA repair mechanism which is activated when DSBs occur or in case of replication fork collapse (Figure 1). Since it requires the presence of sister chromatids to use as templates to repair DNA damage, this pathway generally performs during the S/G2 phase. This system basically involves damage sensors (kinases such as ATM and ATR), signal mediators and facilitators (CHEK1/2, BRIP1), and actual effectors (BRCA1, PALB2, BRCA2, and RAD51) [23]. More specifically, the MRN complex, formed by Mre11, Rad50, and Nbs1, recognizes DSB, and it is responsible for the initiation of DNA end resection from 5′ to 3′. The result of this process is the formation of a single-strand DNA (ssDNA) at the extremity of the DSB, whose degradation is prevented by the attachment of the replication protein A (RPA) [27]. MRN is also crucial for recruiting and activating ataxia telangiectasia mutated (ATM) [28], which cooperates with RPA, leading to the activation of ataxia telangiectasia and Rad-3-related (ATR). Both the protein kinases ATM and ATR proceed to phosphorylate other proteins such as checkpoint kinases 1/2 (CHEK1/2), inducing cell cycle arrest and, on the other hand, assuring signal transduction through the activation of breast cancer 1 (BRCA1). Once phosphorylated by ATM and monoubiquitinated by Fanconi anemia complementation (FANC), even Fanconi anemia complementation group D2 (FANCD2) contributes to the activation of the multi-functional enzyme BRCA1. The complex resulting from the interaction of BRCA1 with BRCA1-associated RING domain 1 (BARD1) contributes to DNA resection, while the association of BRCA1 with BRCA1-interacting protein c-terminal helicase 1 (BRIP1) facilitates DNA repair during replication [17,29]. The coiled-coil domain of BRCA1 mediates its interaction with the partner and localizer of BRCA2 (PALB2) which allows the recruitment of breast cancer 2 (BRCA2) [23,30,31,32]. The formed complex BRCA1-PALB2-BRCA2 finally removes RPA and promotes the assembly of the RAD51 recombinase nucleoprotein filament. The RAD51 recombinase nucleoprotein filament is the actual effector of the invasion of ssDNA into the undamaged sister chromatid to find the homologous sequence for DNA polymerase to use as a template for DNA synthesis [17,33].

Therefore, anything interfering with this multi-step pathway at any level, such as P/LPVs in genes encoding for proteins involved in HRR, makes this meticulous DNA repair mechanism not practicable. This forces cells to rely on the more error-prone non-homologous end-joining (NHEJ) system, inevitably leading to genomic instability and tumorigenesis [24,26].

## 3. Identification of HRD in PC

In the current clinical practice, HRR status in patients with PC could be determined by analyzing tumor tissue or blood samples. Both somatic and germline P/LPVs in HRR-related genes can be detected through DNA extracted from tumor tissue. Once a P/LPV is identified, the analysis of DNA extracted from leukocytes obtained by a peripheral blood sample rules out that the P/LPV is only somatic. The identification of a germline P/LPV implies the need of an appropriate genetic counselling [34].

Tumor tissue is typically acquired through an invasive procedure like biopsy or surgery from primary PC or metastatic sites. In most cases, the tissue sample derives from primary PC core biopsies or surgery; however, it may not reflect the tumor heterogeneity of the metastatic disease. With bone often serving as the unique site of metastasis, it is generally difficult to obtain a tumor sample from a metastatic site in PC. Finally, the cancer tissue may not be suitable for molecular tests due to the small amount and the poor quality of tumoral DNA extracted, mainly due to the progressive degradation in a dated sample, as well as for the presence of inflammatory cells or necrosis [35].

To overcome these limits, liquid biopsy and especially cell-free DNA (cfDNA) detection are emerging techniques in cancer patients’ management. cfDNA consists of circulating DNA released into the blood by cancerous and normal cells. This test can take a picture of the tumor heterogeneity at a given time, identifying both germline and somatic P/LPVs as well as secondary mutations. Nevertheless, this assay has many constraints such as the need of an adequate circulating tumor DNA (ctDNA) level for a reliable result. Currently, the role cfDNA to determine HRR status in PC remains dubious [35].

Up to now, three tests have been approved to determine HRR status in PC and the consequent indication for a therapy with a PARPi. The comprehensive genomic profiling test FoundationOne CDx can detect copy number alterations, substitutions, insertions, and deletions in 324 genes (including HRR-related genes), analyzing DNA extracted from formalin-fixed paraffin-embedded (FFPE) tumor samples. In case of PC, FoundationOne CDx is an FDA-approved companion diagnostic used to detect P/LPVs in *BRCA1*, *BRCA2*, *ATM*, *BARD1*, *BRIP1*, *CDK12*, *CHEK1*, *CHEK2*, *FANCL*, *PALB2*, *RAD51B*, *RAD51C*, *RAD51D,* and *RAD54L* [36,37]. Also, FoundationOne Liquid CDx is a comprehensive genomic profiling test, but this assay isolates cfDNA from peripheral whole blood, and it is an FDA-approved companion diagnostic used to detect *BRCA1*, *BRCA2,* and *ATM* P/LPVs in PC [38]. Finally, Myriad BRACAnalysis CDx is an FDA-approved test used to identify germline P/LPVs in *BRCA1* and *BRCA2* from whole blood samples [39]. Differently from ovarian cancer, genomic scar assays have not been evaluated or approved for the use of PARPis in PC.

## 4. PARPis as Single Agents in Metastatic PC

PARPis have firstly demonstrated their efficacy in patients with mCRPC and HRD. The multicenter randomized phase II study TOPARP-B [40] evaluated the association between P/LPVs in DDR-related genes and response to olaparib in mCRPC patients progressing to at least a taxane-based chemotherapy regimen. A total of 98 patients were randomized to receive 300 or 400 mg of olaparib twice daily. The cohort receiving 400 mg reported higher rates of composite responses defined by the presence of at least one of the following: radiological objective response; a decrease in PSA levels of 50% or more; and the conversion of a circulating tumor cell count (from ≥5 cells to <5 cells per 7.5 mL blood). The highest number of composite responses was observed in the *BRCA1/2* subgroup (83.3%) followed by the *PALB2* (57.1%) and *ATM* (36.8%) subgroups.

On the basis of the TOPARP-B trial, the phase III PROfound trial [41] demonstrated the superiority of olaparib over an androgen receptor signaling inhibitor (ARSI, enzalutamide or abiraterone) not received before in patients with mCRPC progressing to abiraterone or enzalutamide with at least one P/LPV in one of 15 genes selected for their direct or indirect role in HRR (*BRCA1*, *BRCA2*, *ATM*, *BRIP1*, *BARD1*, *CDK12*, *CHEK1*, *CHEK2*, *FANCL*, *PALB2*, *PPP2R2A*, *RAD51B*, *RAD51C*, *RAD51D*, and *RAD54L*). Tumor testing was performed on FFPE tumor samples using FoundationOne CDx. Carriers of P/LPVs in *BRCA1/2* or *ATM* were assigned to cohort A (245 patients) and the rest to cohort B (142 patients). The study met its primary endpoint: the median radiological progression-free survival (rPFS) was significantly longer in the olaparib arm than in the control arm in cohort A (7.4 vs. 3.6 months, hazard ratio (HR): 0.34, 95% confidence interval (CI): 0.25–0.47, *p* < 0.001). Also, in the overall population (cohorts A and B), the median rPFS was significantly longer in the olaparib arm than in the control arm (5.8 vs. 3.5 months, HR: 0.49, 95%CI: 0.38–0.63, *p* < 0.001). Despite the fact that 66% of the patients belonging to the control group crossed over to olaparib after radiological disease progression, the median overall survival (mOS) was significantly longer in the olaparib arm than in the control arm in cohort A (19.1 vs. 14.7 months, HR: 0.69, 95%CI: 0.50–0.97, *p* = 0.02). Exploratory analyses revealed the maximum benefit in terms of OS in the presence of *BRCA1* (HR: 0.42, 95%CI: 0.12–1.53) or *BRCA2* (HR: 0.59, 95%CI: 0.37–0.95) P/LPVs, while the HR for death among patients with a P/LPV in any non-*BRCA* gene was 0.95 (95%CI: 0.68–1.34) [42]. The main criticism of this study is the treatment employed in the control arm: the use of an ARSI after a previous therapy with an ARSI is associated with a limited efficacy in most patients with mCRPC for the development of cross-resistance [43,44,45,46], and a taxane-based chemotherapy would have been probably a better control arm. Moreover, the olaparib benefit in cohort A was driven by *BRCA1/2* P/LPVs carriers: the HR for death among patients with a P/LPV in *ATM* was 0.93 (95%CI: 0.53–1.75), similar to what was observed in carriers of P/LPV in any non-*BRCA* gene.

The PARPi rucaparib was firstly tested in the phase II TRITON2 [47] and then in the phase III TRITON3 trial [48]. TRITON2 enrolled a total of 277 patients with mCRPC progressing to at least an ARSI and one taxane-based chemotherapy harboring at least one P/LPV in one of the 15 selected DDR-related genes (*BRCA1*, *BRCA2*, *ATM*, *CDK12*, *CHEK2*, *PALB2*, *FANCA*, *BARD1*, *BRIP1*, *NBN*, *RAD51*, *RAD51B*, *RAD51C*, *RAD51D,* and *RAD54L*) detected through the genomic profiling of either cfDNA or FFPE tumor tissue by Foundation Medicine, Inc. or by local testing. After a median follow-up of 23.7 months in the *BRCA1/2* subgroup and 25.8 months in the other genes subgroup, rucaparib given 600 mg twice daily resulted in an objective response rate (ORR) of 46% in patients with *BRCA1*/*2* P/LPVs, while no patient with *ATM*, *CDK12*, or *CHEK2* P/LPV had an ORR. The ORR was 100% in *PALB2* P/LPV carriers (only 4 patients) and 25% in the other genes subgroup (including only 12 patients) [47].

The phase III TRITON3 trial randomized a total of 405 mCRPC patients harboring at least one P/LPV in *BRCA1*/*2* or *ATM*, identified through the same modalities of TRITON2, and progressing to a previous ARSI to receive rucaparib or the physician’s choice treatment which could include docetaxel or an ARSI (abiraterone or enzalutamide). Among the control group, 56% of the patients received docetaxel. After a follow-up of 62 months, the median rPFS was significantly longer in the rucaparib arm than in the control arm in both *BRCA1/2* P/LPV carriers (11.2 vs. 6.4 months, HR: 0.50, 95%CI: 0.36–0.69, *p* < 0.001) and in the intention-to-treat population (10.2 vs. 6.4 months, HR: 0.61, 95%CI: 0.47–0.80, *p* < 0.001); however, an exploratory analysis of the *ATM* subgroup revealed an rPFS similar in the rucaparib and control arms (8.1 vs. 6.8 months, HR: 0.95, 95%CI: 0.59–1.52). The data regarding OS were immature, although a trend in favor of rucaparib was reported [48].

All the abovementioned studies agree that the greatest effectiveness of PARPis is observed in the presence of *BRCA1*/*2* P/LPVs, while the efficacy in the case of P/LPVs in other genes remains uncertain. The rarity of P/LPVs in genes other than *BRCA2* in PC makes indeed for difficult subgroups gene analyses. Furthermore, the majority of the genes analyzed encode for proteins not directly involved in HRR: for example, ATM is a damage sensor, whose role could be partially replaced by ATR [49], differently from BRCA1, BRCA2, and PALB2, which are direct effectors.

## 5. Crosstalk between HRR and Androgen Receptor Pathway

Even in the presence of HRD, PC remains an androgen-dependent tumor [50] whose receptor (AR) plays a critical role in PC pathogenesis and progression [51]. This provides the rationale for PC treatment with ADT and an ARSI. In detail, testosterone induces tumor growth and progression through both a non-genomic and genomic signaling pathway. Testosterone is metabolized to 5α-dihydrotestosterone (DHT) by the enzyme 5α-reductase. DHT exerts its biological effects by binding AR in the cytoplasm. A rapid non-genomic signaling pathway, initiated by the association of AR with molecular substrates into the cytoplasm, contributes to cell proliferation by the activation of the MAPR/ERK and PI3K/AKT pathways and by the exclusion of other steroid receptors from the nucleus [52,53,54]. In addition, once it has bound the ligand, AR homodimerizes and translocates into the nucleus, where it binds to the androgen response element (ARE) located at the promoter regions of genes involved in cell proliferation and apoptosis evasion [55,56]. In this way, the AR pathway promotes DDR response, including HRR, to guarantee genome integrity during DNA replication (Figure 1).

The inhibition of AR signaling consequently leads to the downregulation of HRR-related genes expression, which in turn results in the accumulation of DNA damage. DNA damage, in particular DNA SSBs, activates PARP signaling. In presence of PARP inhibition, DNA SSBs cannot be repaired, with a consequent replication fork collapse during DNA replication and the further accumulation of DNA damage, which ultimately leads to cell death, which is likely to happen in any androgen-independent tumor [17,57,58]. In addition, PARP plays a role in androgen-dependent transcription in an NAD-independent manner, so PARP inhibition may in turn impair AR signaling [59]. This crosstalk between the AR pathway and DDR processes has paved the way towards new therapeutic strategies based on the combination of ADT and an ARSI with PARPis also in tumors without HRD. Clinical trials in this field will be described in the following section.

## 6. Therapy Combinations with PARPis in Metastatic PC

In patients with mCRPC, several therapy combinations of PARPis with an ARSI have been evaluated in different phase III studies over the last years, regardless of HRR status. Up to now, three randomized, double-blinded, placebo-controlled studies have reported positive results: PROpel, TALAPRO-2, and MAGNITUDE.

PROpel [60] evaluated the addition of olaparib to abiraterone as the first-line treatment of mCRPC compared to abiraterone plus a placebo. The study met its primary endpoint: after a median follow-up of 19.3 months, the median rPFS was 24.8 months in the experimental arm compared to 16.6 months in the control group (HR: 0.66, 95%CI: 0.54–0.81, *p* < 0.001). In the HRD group, the median rPFS had not yet been reached. However, the study did not meet its secondary endpoint: in the final prespecified analysis after a follow-up of 36.5 months [61], mOS did not statistically differ between the two treatment arms (42.1 vs. 34.7 months, HR: 0.81, 95%CI: 0.67–1.00, *p* = 0.054). Despite this, a post hoc exploratory subgroups analysis revealed that the treatment with abiraterone and olaparib reduced the risk of death by 71% of in the *BRCA1/2* subgroup (mOS: not reached vs. 23.0 months, HR: 0.29, 95%CI: 0.14–0.56) and by 34% in the HRD subgroup (mOS: not reached vs. 28.5 months, HR: 0.66, 95%CI: 0.45–0.95).

TALAPRO-2 [62] compared the combination of talazoparib and enzalutamide with enzalutamide plus a placebo as the first-line treatment of mCRPC. The addition of talazoparib resulted in a statistically significant improvement of rPFS: in the planned primary analysis after a median follow-up of 24.6 months, the median rPFS was not reached in patients treated with talazoparib plus enzalutamide versus 21.9 months in the control group (HR: 0.63, 95%CI: 0.51–0.78, *p* < 0.0001). The maximum benefit from the combination therapy was observed in patients with HRD (median rPFS: 27.9 vs. 16.4 months, HR: 0.46, 95%CI: 0.30–0.70, *p* = 0.0003). Differently from PROpel where HRR status was determined following enrollment, in TALAPRO-2, the randomization was stratified according to HRR status. Survival follow-up is ongoing and will further elucidate the benefit of this combination in both HRD- and HRR-proficient tumors.

MAGNITUDE [57] assessed niraparib plus abiraterone compared to abiraterone plus a placebo as the first-line treatment of mCRPC. In this case, the HRR status was prospectively determined. Also, in this study, the primary endpoint was rPFS, but it was first evaluated in the *BRCA1/2*-mutated cohort and then in the full HRD subgroup. For the HRR-proficient cohort, futility was declared and the enrollment was closed. After a median follow-up of 18.6 months, the median rPFS was significantly longer in patient with either *BRCA1/2* mutated or HRD tumors treated with the therapy combination (*BRCA1/2* subgroup: 16.6 vs. 10.9 months, HR: 0.53, 95%CI: 0.36–0.79, *p* = 0.001; HRD subgroup: 16.5 vs. 13.7 months, HR: 0.73, 95%CI: 0.56–0.96, *p* = 0.022). The OS data were still immature at this first analysis.

In all these studies, toxicity in the combination arms was generally manageable and consistent with the known safety profiles of single drugs. The most commonly reported adverse events coincided with the most well-known frequent side effects related to PARPi treatment: anemia, fatigue, and nausea. A comparison among these three studies is quite difficult (Table 1). First of all, all three PARPis are potent inhibitors of PARP with comparable half-maximal inhibitory concentration values, but they differ in their PARP-trapping potency: talazoparib has a trapping ability 50- to 100-fold higher than niraparib and olaparib. As single agents, the cytotoxicity of PARPis has proven to correlate with PARP trapping and not with PARP catalytic inhibition [63]. Secondly, it is noteworthy that the HRR status was determined with different technologies and gene panels. For PROpel, both tumor tissue and ctDNA-based (FoundationOne CDx) tests were employed to detect P/LPVs in the following genes: *ATM*, *BRCA1*, *BRCA2*, *BARD1*, *BRIP1*, *CDK12*, *CHEK1*, *CHEK2*, *FANCL*, *PALB2*, *RAD51B*, *RAD51C*, *RAD51D*, and *RAD54L* [60]. In TALAPRO-2, HRR status was assessed in tumor tissue, and a subsequent protocol amendment permitted prospective ctDNA testing (FoundationOne CDx); the genes analyzed were *ATM*, *ATR*, *BRCA1*, *BRCA2*, *CDK12*, *CHEK2*, *FANCA*, *MLH1*, *MRE11A*, *NBN*, *PALB2*, and *RAD51C* [62]. In MAGNITUDE, HRR status was determined by testing both tissue and plasma with FoundationOne CDx, the Resolution Bioscience HRD plasma test, or AmoyDx blood and tissue assays; at least one P/LPV among *ATM*, *BRCA1*, *BRCA2*, *BRIP1*, *CDK12*, *CHEK2*, *FANCA*, *HDAC2*, and *PALB2*, detected in at least one assay was needed to be considered HRD [57]. Finally, the durations of follow-up currently available are different among the three studies, and the OS data are lacking for TALAPRO-2 and MAGNITUDE.

## 7. Mechanisms of Resistance to PARPis in PC

The increasing use of PARPis is raising the issue of resistance to therapy. Several different mechanisms of resistance have been proposed, mainly based on breast and ovarian cancer studies and preclinical models, although only the acquisition of secondary mutations in *BRCA1/2* has been clinically proved [16,64]. The restoration of HRR capabilities in cells with HRD may occur by reversion mutations, epigenetic modification, or loss of DNA end resection regulation. Reversion mutations consist of secondary genetic alterations that, by reinstating the open reading frame, lead to the recovery of HRR proficiency. First discovered in *BRCA2* [65], reversions can also occur in *BRCA1*, *PALB2*, *RAD51C* and *RAD51D*, and *ATM* [66,67,68]. In PC, reversion mutations of *BRCA2* have been observed in a small number of mCRPC patients treated with PARPis, including olaparib, or carboplatin [66,67,68,69,70,71,72,73].

Several other mechanisms of resistance to PARPi have been proposed, although they have not been observed in clinical practice up to now (reviewed in [16]). For example, *BRCA1* promoter demethylation, as a consequence of previous multiple lines of treatment, could rescue the expression of BRCA1 and conferred resistance of PARPis [74,75]. The loss of DNA end resection regulation, due to the genetic depletion of *53BP1*, *RIF1*, or *REV7*, provides synthetic viability to *BRCA1-null* cell lines and provides resistance to PARPis, restoring HRR [16]. In addition to the mechanisms of resistance intrinsic to the DDR, also the decreased expression of PARP and PARP point mutations interfering with the DNA-binding domain can cause PARPi resistance and affect PARP trapping, as well as the upregulation of drug efflux pumps [75,76,77,78].

## 8. Ongoing Trials with PARPis in Metastatic PC

The promising therapeutic role of PARPis in patients with metastatic PC needs to be furtherly explored. Several clinical trials are currently ongoing to evaluate the efficacy, safety and tolerability of PARPis in combination with other drugs.

### 8.1. Phase III Trials

Up to now, three phase III studies are evaluating the potential therapeutic synergy between PARPis and an ARSI in advanced settings (Table 2). The enrollment is completed in all these studies. CASPAR (NCT04455750) is the only study still evaluating the benefit of the combination of a PARPi with an ARSI (rucaparib + enzalutamide) as the first-line treatment of mCRPC, regardless of HRR status. The primary endpoints are rPFS and OS in the intention-to-treat population; secondary endpoints include rPFS and OS in patients with *BRCA1/2-* or *ATM*-mutated tumors.

The increasing use of an ARSI with ADT as the first-line treatment of mHSPC, based on the survival benefit demonstrated by several phase III trials [79,80,81], has prompted the question of whether the further addition of a PARPi in this setting in the presence of HRD can result in an even greater benefit. Both TALAPRO-3 (NCT04821622) and AMPLITUDE (NCT04497844) are trying to answer to this question by assessing the efficacy and safety of adding a PARPi to an ARSI as the first-line treatment of mHSPC with HRD. In detail, the experimental arm is enzalutamide + talazoparib in TALAPRO-3 and abiraterone + niraparib in AMPLITUDE. The primary endpoint is rPFS in both studies.

### 8.2. Phase II Trials

Numerous ongoing phase II studies in advance PC are combining PARPis with different drugs other than an ARSI with the double aim of enhancing efficacy and overcoming resistance to PARPi in both HRD and HRR-proficient tumors (Table 3, Figure 1). For example, olaparib (NCT03263650), niraparib (NCT04592237, NCT04288687), and rucaparib (NCT03442556) are being tested as maintenance therapy after chemotherapy, with taxanes (docetaxel or cabazitaxel) routinely used in metastatic PC and/or carboplatin. Carboplatin, like temozolomide (which is tested in association with talazoparib in NCT04019327), is an alkylating agent that induces DNA interstrand crosslinks (ICLs); endonucleases cleave both 3′ and 5′ strand at the ICL site, forming a DSB [82].

Several target agents are being evaluated in combination with a PARPi to further disrupt the DDR machinery. Asi-DNA (NCT05700669) consists of small double-stranded DNA molecules that bind to and activate both PARP and DNA-dependent protein kinase by mimicking DSB, thus triggering an inappropriate genomic repair signal and preventing the recruitment and the activity of enzymes required for HRR and NHEJ at endogenous DNA damage sites [83]. Ceralasertib (NCT03787680, NCT03682289) is an ATR inhibitor. TNG348 (NCT06065059) is an USP1 inhibitor; USP1 is a critical regulator of ICL repair and HRR, being required, among other things, for the deubiquitination of FANCD2-Ub [84].

Other drugs under investigation are agents targeting cellular pathways involved in tumor growth and progression. NUV-868 (NCT05252390) is a BRD4 inhibitor; BRD4 is a transcriptional coactivator of genes involved in tumor progression, angiogenesis, metastasis, and resistance to therapies [85]. CCS1477 (NCT03568656) is an inhibitor of p300 and CBP, which are two closely related transcriptional activators of AR [86]. Copanlisib (NCT04253262) is a pan-PI3K inhibitor. Cediranib (NCT02893917) is a VEGFR inhibitor.

Finally, using PARPis and immune checkpoints inhibitors (ICIs) is a rational combination: PARPi-induced DNA damages increase genomic instability, immune pathway activation, and PD-L1 expression on cancer cells which might promote responsiveness to ICIs [87]. Numerous phase II trials are testing this hypothesis in advanced PC (NCT05005728, NCT03682289, NCT02484404, and NCT04592237).

## 9. Conclusions

The introduction of PARPis is rapidly changing the standard of care of patients affected by mCRPC with P/LPVs in *BRCA1/2*. The full potential of PARPi therapy in mHSPC as well as in the case of P/LPVs in other HRR-related genes needs to be furtherly explored. In HRR-proficient tumors, the possibility of combination therapies including a PARPi is an attractive option, but more robust data are awaited from ongoing phase II and phase III trials.

## Figures and Tables

**Figure 1 ijms-25-04624-f001:**
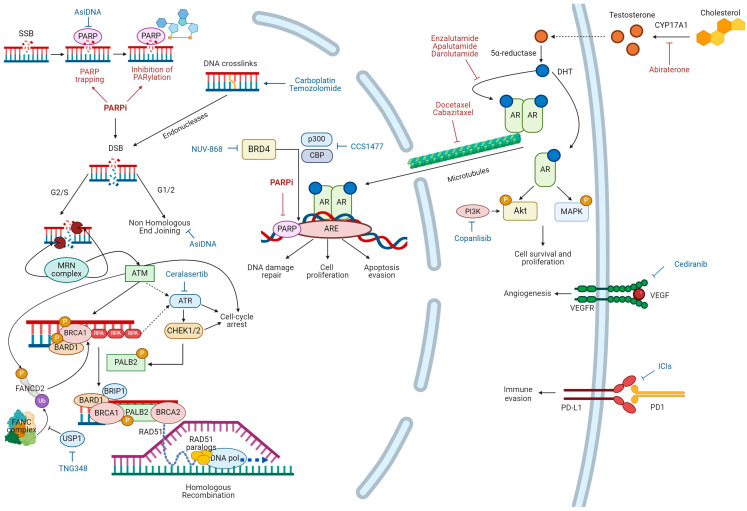
When a DNA single-strand break (SSB) happens, poly (ADP-ribose) polymerase (PARP) guides SSB repair through NAD^+^poly(ADP-ribosyl)ation (PARylation) of histones and chromatin remodeling enzymes. PARP inhibitors (PARPis) act mainly by trapping PARP and by inhibiting PARylation. Unrepaired SSB leads to DNA double-strand break (DSB) during DNA replication. DSB is mainly resolved by homologous recombination repair (HRR) during the G2/S phase of cell cycle. During G1/G2 and in HRR deficient cells, DSB is solved by the error-prone non-homologous end-joining pathway with consequent genomic instability and cell death. HRR is activated by the recognition of DSB by the MRN complex that resects DNA ends, leading to the formation of single-strand DNA (ssDNA). ssDNA is preserved from degradation by RPA. The MRN complex activates ATM and that with RPA contributes to ATR activation. ATM and ATR phosphorylate several proteins such as CHEK1/2. ATM, ATR, and CHEK1/2 mediate cell cycle arrest. FANCD2 contributes to BRCA1 activation once monoubiquitinated by FANC and phosphorylated by ATM. BRCA1-BARD1 complex facilitates DNA end resection and interacts with the bridging protein PALB2 phosphorylated by CHEK2. PALB2 recruits BRCA2. PALB2 and BRCA2 remove RPA and facilitate the assembly of RAD51. RAD51 and RAD51 paralogs mediate strand invasion of ssDNA into the intact sister chromatid, searching a homologous template for DNA synthesis by DNA polymerase (DNA pol). DNA repair process interacts with androgen receptor (AR) pathway. CYP17A1 is a key enzyme for testosterone synthesis from cholesterol. Testosterone is metabolized to 5α-dihydrotestosterone (DHT) by 5α-reductase. DHT exerts its biological effects by binding AR in the cytoplasm. A rapid non-genomic signaling pathway contributes to cell proliferation by activation of MAPR/ERK and PI3K/AKT pathways. In addition, once it has bound the ligand, AR homodimerizes and translocates into the nucleus, where it binds to the androgen response element (ARE) located at the promoter regions of genes involved in DNA damage repair, cell proliferation and apoptosis evasion. PARP plays also a role in androgen-dependent transcription. Drugs currently used in clinical practice alone or in combination with PARPis for the treatment of prostate cancer are labelled in red. Abiraterone is a CYP17A1 inhibitor. Enzalutamide, apalutamide, and darolutamide competitively inhibit DHT binding to the AR, nuclear translocation of the AR, and DNA binding. Docetaxel and cabazitaxel inhibit AR nuclear translocation by targeting AR association with microtubules. Drugs that are currently being explored in combination with PARPis are labelled in blue.

**Table 1 ijms-25-04624-t001:** Phase III studies evaluating therapy combinations of PARPis with ARSI in mCRPC.

Study	Treatment Arms	Median Follow-Up	Median rPFS in ITT Population	HR (95%CI)	Median rPFS in HRD Group	HR (95%CI)	Tests to Determine HRR Status
PROpel	Abiraterone + olaparib Abiraterone + placebo	19.3 mo 19.4 mo	24.8 mo 16.6 mo	0.66 (0.54–0.81)	NR 13.9 mo	0.5 (0.34–0.73)	Tissue and ctDNA (FoundationOne CDx)
TALAPRO-2	Enzalutamide + talazoparib Enzalutamide + placebo	24.9 mo 24.6 mo	NR 21.9 mo	0.63 (0.51–0.78)	27.9 mo 16.4 mo	0.46 (0.30–0.70)	Tissue and ctDNA (FoundationOne CDx)
MAGNITUDE *	Enzalutamide + niraparib Enzalutamide + placebo	18.6 mo 18.6 mo	- -	- -	16.5 mo 13.7 mo	0.73 (0.56–0.96)	Tissue and ctDNA (FoundationOne CDx, Resolution HRD, AmoyDx)

* HRR-proficient cohort closed for futility. Moreover, 95%CI: 95% confidence interval; HR: hazard ratio; HRD: homologous recombination deficiency; HRR: homologous recombination repair; ITT; intention to treat; rPFS: radiological progression-free survival.

**Table 2 ijms-25-04624-t002:** Ongoing phase III studies evaluating therapy combinations of PARPi with ARSI.

Official Title NCT Number	Setting	Experimental Arm	Control Arm	Primary Endpoints	Status	Enrolment	Primary Completion
CASPAR NCT04455750	I-line mCRPC, regardless of HRR status	Enzalutamide + rucaparib	Enzalutamide + placebo	rPFS, OS	Active, not recruiting	61 (actual)	May 2024
TALAPRO-3 NCT04821622	I-line mHSPC with HRD	Enzalutamide + talazoparib	Enzalutamide + placebo	rPFS	Active, not recruiting	599 (actual)	September 2025
AMPLITUDE NCT04497844	I-line mHSPC with HRD	Abiraterone + niraparib	Abiraterone + placebo	rPFS	Active, not recruiting	696 (actual)	November 2024

HRD: homologous recombination deficiency; HRR: homologous recombination repair; mHSPC, metastatic hormone-sensitive prostate cancer; mCRPC: metastatic castration-resistant prostate cancer; OS: overall survival; rPFS: radiological progression-free survival.

**Table 3 ijms-25-04624-t003:** Ongoing phase II studies evaluating PARPi in metastatic PC.

NCT Number	Setting	Treatment	Primary Endpoints	Status	Enrolment	Primary Completion
NCT05501548	≥II-line mCRPC HRR-proficient	Olaparib + ascorbate	PSA50	Recruiting	15 (estimated)	March 2028
TRAP trial NCT03787680	≥II-line mCRPC, regardless of HRR status	Olaparib + ceralasertib	ORR in HRR-proficient patients	Active, not recruiting	49 (actual)	January 2023
FAALCON NCT04748042	Oligometastatic HSPC	Olaparib + abiraterone + radiotherapy	Percentage of patients without treatment failure at 24 months	Recruiting	29 (estimated)	May 2025
NCT03263650	Aggressive variants of metastatic PC	Olaparib maintenance after six cycles of cabazitaxel and carboplatin	PFS	Active, not recruiting	119 (actual)	June 2024
NCT02893917	>II-line mCRPC	Olaparib ± cediranib	rPFS	Active, not recruiting	90 (estimated)	December 2023
NCT05167175	I-line mHSPC with HRD	Olaparib + abiraterone	rPFS	Recruiting	30 (estimated)	December 2024
NCT05005728 Cohort C	≥II-line mCRPC with HRD/CDK12 biallelic loss tumors	Olaparib + vudalimab	Incidence of treatment-related AEs	Recruiting	85 * (estimated)	June 2023
NCT05700669 mCRPC cohort	mCRPC progressed to previous PARPi	Olaparib + AsiDNA	ORR	Recruiting	115 * (estimated)	December 2026
NCT05252390 mCRPC cohort	mCRPC progressed to a previous ARSI	NUV-868 ± olaparib or enzalutamide	ORR, PSA50, rPFS	Recruiting	657 * (estimated)	June 2026
NCT03682289 Cohort D	mCRPC progressed to a previous ARSI	Ceralasertib ± olaparib or durvalumab	ORR, PSA50	Recruiting	89 * (estimated)	July 2025
NCT06065059 PC cohort	Metastatic PC *BRCA1/2* mutant or with HRD	TNG348 ± Olaparib	ORR	Recruiting	140 * (estimated)	December 2025
NCT03568656 mCRPC cohort	mCRPC progressed to a previous ARSI and docetaxel	CCS1477 ± olaparib or abiraterone or enzalutamide or darolutamide	Incidence of treatment-related AEs, laboratory assessments	Recruiting	350 * (estimated)	March 2024
NCT02484404 Cohort 4	mCRPC progressed to a previous ARSI and/or docetaxel	Olaparib + durvalumab	ORR, AEs, PSA response	Recruiting	384 * (estimated)	December 2024
NCT04332744	I-line high-volume mHSPC	Enzalutamide ± talazoparib	PSA-CR	Active, not recruiting	54 (actual)	April 2025
NCT04734730	I-line mHSPC	Talazoparib + abiraterone	PSA-CR	Recruiting	70 (estimated)	August 2027
NCT04019327	mCRPC without DDR mutations progressed to a previous ARSI	Talazoparib + temozolomide	ORR	Recruiting	44 (estimated)	July 2027
NCT04550494 PC cohort	Metastatic PC with DDR mutations	Talazoparib	Rate of patients with Rad51 activation	Recruiting	30 * (estimated)	December 2024
KNIGHTS NCT06212583	Recurrent oligometastatic HSPC with high-risk DDR mutations	Radiotherapy ± niraparib and abiraterone	PSA at the 18-month progression	Not yet recruiting	88 (estimated)	December 2028
NCT04592237	Aggressive variants of metastatic PC	Niraparib ± cetrelimab maintenance after six cycles of cabazitaxel + carboplatin + cetrelimab	PFS	Recruiting	120 (estimated)	December 2025
NCT05689021	mCRPC with *SPOP* mutations progressed to a previous ARSI	Niraparib + abiraterone	PSA50	Recruiting	30 (estimated)	September 2024
PLATPARP NCT04288687	Platinum-sensitive mCRPC with DDR mutations	Niraparib maintenance	6-month rPFS	Active, not recruiting	12 (actual)	June 2024
TRIUMPH NCT03413995	mHSPC with HRD in patients refusing ADT	Rucaparib	PSA50	Active, not recruiting	30 (estimated)	November 2023
NCT04253262	mCRPC progressed to a previous ARSI with HRD	Rucaparib + Copanlisib	ORR	Active, not recruiting	13 (actual)	January 2024
PLATI-PARP NCT03442556	mCRPC with HRD	Rucaparib maintenance after 4 cycles of docetaxel and carboplatin	rPFS	Active, not recruiting	18 (actual)	May 2025

* Including all the cohorts in the study. ADT: androgen deprivation therapy; AEs: adverse events; ARSI: androgen receptor signaling inhibitor; DDR: DNA damage repair; HRD: homologous recombination deficiency; HRR: homologous recombination repair; mCRPC: metastatic castration-resistant prostate cancer; mHSPC, metastatic hormone-sensitive prostate cancer; ORR: overall response rate; PSA50: PSA decrease > 50% compared to baseline; PSA-CR: PSA complete response (≤0.3 ng/mL); rPFS: radiological progression-free survival.

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
