# Peer review of "Homologous Recombination Repair Deficiency in Metastatic Prostate Cancer: New Therapeutic Opportunities"

_ijms, 2024, doi:10.3390/ijms25094624_

Round 1
Reviewer 1 Report
Comments and Suggestions for Authors
The review titled “Homologous Recombination Repair Deficiency in Metastatic Prostate Cancer: New Therapeutic Opportunities” is well written. However, I have a few following comments.
1. Page1- Line 39, the authors must indicate the subgroup of patients in question
2. Page 1- Line 44, explain briefly how hormone sensitive phase leads to heterogeneity,
3. Figure 1 legend should be made less descriptive.
4. Have cancers reported any resistance against Olaparib?
5. Page 2, line 7 with is misspelled.
6. The authors should make sure that there are no spelling errors throughout the manuscript.
7. Are there any studies in PCs evaluating the effect of other PARPi inhibitors?
Author Response
- Done
- Done
- Figure 1 legend has been shortened.
- PC resistance against olaparib has been reported as specified in the paragraph 7
- Fixed
- The manuscript has been reviewed to fix spelling errors
- Studies in PCs evaluating the effect of other PARPi inhibitors have been extensively described in paragraphs 4, 6, and 8.
Reviewer 2 Report
Comments and Suggestions for Authors
The authors present a thorough revision of the current therapeutic approaches for Prostate Cancer considering patients’ eligibility to PARP inhibitors and the signalling pathways being targeted by the different therapeutic approaches under study. The content of the review is interesting and highly updated.
The compilation of all signalling pathways and alternative therapeutic approaches in a single figure, constantly being cited throughout the manuscript, does not work.
Comments on the Quality of English LanguageOverall, the manuscript is well structured and the content in each section is interesting. However, the English language is careless, making the reading hard and heavy. On one hand, it seems to be written in "spoken English" to be easily understandable by the general population, but at the same time, it uses heavy molecular language without basic explanations, making the reading hard to follow.
Author Response
The choice of including a single figure with all signalling pathways and alternative therapeutic approaches was made to represent the idea of network and complexity behind the new therapeutic opportunities in prostate cancer.
The manuscript was submitted for the special issue Molecular Research on Prostate Cancer that "aims to consolidate the latest findings and perspectives on the molecular landscape of prostate cancer, genomic alterations, signaling pathways and novel biomarkers". Therefore, from this point of view, a "heavy molecular language" is required, even without basic explanations, since the IJMS "provides an advanced forum for molecular studies in biology and chemistry, with a strong emphasis on molecular biology and molecular medicine".
Reviewer 3 Report
Comments and Suggestions for Authors
The manuscript tries to give an answers to these questions: whether PARPi are effective for all patients with HRD; how to identified HRD in PC; the role of PARPi in patients with HRR proficient PC; and the future of the precision medicine in metastatic PC.
Overall, the paper is well organized and its presentation is clear. The following are the questions and some mistakes in this manuscript:
1) HRD is supposed to be responsive to PARP inhibitors. The effectiveness of PARP inhibitors is observed in presence of BRCA1/2 P/LPVS. Whether are PARPi effective in case of P/LPVs in other genes in metastatic PC? Is there any literature report?
2) In line 282 on page 7, the authors mentioned that “PARP plays a role in androgen-dependent transcription in a NAD-independent manner”. What is the effectiveness of PARP inhibitors in androgen-independent tumor?
3) In presence of PARP inhibition, DNA SSBs cannot be repaired, with consequent replication fork collapse during DNA replication and further accumulation of DNA damage, which ultimately leads to cell death. What are the side effects of PARP inhibitors treatment?
Author Response
- PARPi effectiveness in case of P/LPVs in other genes than BRCA1/2 in metastatic PC was discussed for each clinical trial described in the paragraph 4 and 6.
- The effectiveness of PARPi in androgen-independent tumors (although only few PC can be defined androgen-independent in the latter stages of the disease) relies on its ability to impair HRR as described in paragraph 2. A further clarification has been added.
- The most common side effects of PARPi are reported in paragraph 6.